

# Flow Intake Control using Dry-weather Forecast

Otto Icke[1], Kim van Schagen[1], Christian Huising[2], Jasper Wuister[1], Edward van Dijk[1], Arjan Budding[2]

[1]Business Unit Water, Royal HaskoningDHV, Amersfoort, 3800 BC, the Netherlands
[2]Policy Department, Water authority Vallei en Veluwe. Apeldoorn, 7320 AC, the Netherlands

*Correspondence to*: Otto Icke (otto.icke@rhdhv.com)

**Abstract.** Level-based control of the influent flow causes peak discharges at a waste water treatment plant (WTTP) after rainfall events. Furthermore, the capacity of the post-treatment is in general smaller than the maximum hydraulic capacity of the WWTP. This results in a significant bypass of the post-treatment during peak discharge. The optimisation of influent flow reduces peak discharge, and increases the treatment efficiency of the whole water cycle, which benefits the surface
water quality. In this paper, it is shown that half of the bypasses of the post-treatment can be prevented by predictive control. A predictive controller for influent flow is implemented using the Aquasuite® Advanced Monitoring and Control platform. Based on real-time measured water levels in the sewerage and both rainfall and dry-weather flow (DWF) predictions, a discharge limitation is determined by a volume optimisation technique. For the analysed period (February – September 2016) results at WWTP Bennekom show that about 50% of bypass volume can be prevented. Analysis of single rainfall
events shows that the used approach is still conservative and that the bypass can be even further decreased by allowing discharge limitation during precipitation.

## 1 Introduction

The influent flow to most waste water treatment plants (WWTPs) is currently controlled by only the level in the sewerage. This level-based control of the influent flow causes peak discharges at a WTTP after rainfall events. Even if there is enough
storage available in the sewerage to discharge the precipitation in a more gradual way. As a result both hydraulic and biological peak loads are considerable. This affects the performance of the WWTP. Furthermore, the capacity of the post-treatment is in general smaller than the maximum hydraulic capacity of the WWTP. This results in a significant bypass of the post-treatment during peak discharge.

Therefore, reduction of peak discharge by optimising the influent flow control is expected to be effective. Predictive control with dry-weather forecast can be applied to limit the influent flow at the end of (or even during) rainfall events. Where level-based control is characterised by a reactive response, predictive control anticipates on changing circumstances. In this way the available storage of the sewerage can be used without causing extra combined sewer overflow (CSO) or water on street (WOS). This increases the treatment efficiency of the whole water cycle, which benefits the surface water quality.




For WWTP Woudenberg preliminary study was carried out on using predictive control. It was shown that the amount of bypass can be reduced by approximately 65% (van Dijk 2013). The study used a conservative approach, in which the discharge is limited *after* the rainfall event. It was suggested that the amount of bypass could be even more reduced in case the discharge is limited *during* the rainfall event. However this could lead to more CSO or WOS if amounts of rainfall are
predicted wrong. To verify the preliminary study, a pilot project was started for four catchment areas, to investigate the true reduction in bypass using predictive control.

## 2 Material and methods

A predictive controller for influent flow is implemented in this pilot project using the Aquasuite® Advanced Monitoring and Control platform (van der Kolk 2016). Rainfall and dry-weather flow (DWF) predictions are applied as the basis for the
predictive control. The predictive control is used to limit the discharge to the treatment plant.

The rainfall prediction is obtained from the High Resolution Limited Area Model (HIRLAM) of the Royal Netherlands Meteorological Institute (KNMI). The DWF prediction is obtained by a measurement data driven technique (Bakker at al. 2013a). Based on real-time measured water levels in the sewerage and both rainfall and DWF predictions, a discharge
limitation is determined by a volume optimisation technique (Bakker et al. 2013b). This is shown in Figure 1.

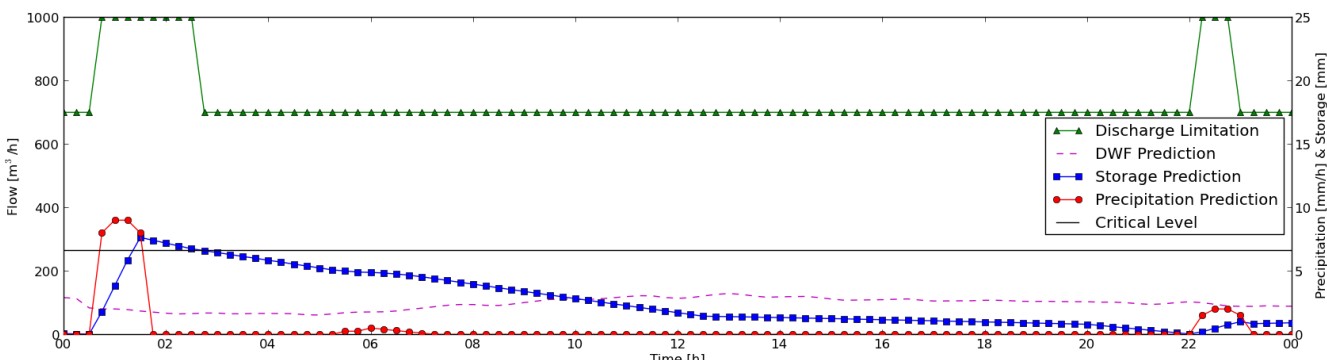

**Figure 1: Outline of the predictive control for the influent flow of WWTP Bennekom (Icke et al. 2016).**


During a rainfall event, the discharge limitation is at maximum capacity. After the rainfall event, when levels are below a defined critical level, the sewerage is emptied with a limited discharge. In Figure 1 an example with a discharge limitation at 70% of the maximum capacity is shown. Based on the time until the next rainfall event and the maximum time to empty the sewerage, the influent flow is optimised between the maximum capacity and the capacity of the post-treatment.




## 2.1 Rainfall forecast

As travelling times of the sewerage can take up to 24 hours (RIONED 2008), precipitation predictions with similar periods are required. Optimal utilisation of the storage of the sewerage is only possible with information of subsequent rainfall peaks. Therefore, at the beginning of this project, HIRLAM of the KNMI was selected as the numerical weather prediction

(NWP) forecast system from which the precipitation data was obtained. This rainfall forecast has raster cell sizes of 11,0 by 7,0 km, a forecast horizon of 48 hours and a refresh rate of circa 6 hours. A single time series is obtained for each specified area by transforming the information of the raster cells within the polygon by application of a geostatistical method.

## 2.2 Dry-weather flow prediction

An accurate prediction of the dry-weather flow is essential to determine the total predicted flow to the WWTP. The influent

flow has, at WWTPs with mainly domestic waste water, a typical day pattern. Both (time-dependent) water demand of inhabitants and contribution of industrial discharges, as well as properties of the sewerage like travelling time influence the exact characteristics of this pattern.

A data driven technique is applied to obtain this DWF prediction which is self-learning based on real-time flow

measurements. This particular technique is based on the fully adaptive forecasting model for short-term drinking water demand (Bakker et al. 2013a) applied on waste water discharge. Before this pilot project, the application of this technique on waste water discharge had already successfully been implemented, yet with a rather different objective. DWF prediction is an accurate measure for the load of biological oxygen demand (BOD) prediction. Therefore, it can be applied within a predictive controller for aeration (de Koning et al. 2013).


Based on previously occurring patterns of each specific day of the week, a daily curve is automatically obtained. However, except for those with deviations due to peak flows caused by precipitation. Combination of the prediction of the total daily volume with this predicted curve determines a DWF prediction. For each specific catchment area, a distinct DWF prediction is obtained. The DWF prediction has a 48 hours forecast horizon, just like the rainfall forecast.

## 2.3 Predictive control

A volume optimisation technique is applied to determine predicted storage and discharge. Together with real-time measured water levels and boundary conditions, both DWF and precipitation predictions are used as an input. Firstly, actual utilised storage of the sewerage is derived from real-time water level measurements and relationships between level and volume. These relationships are obtained from storage curves derived from sewer models and are verified with measured values for

level and discharge.

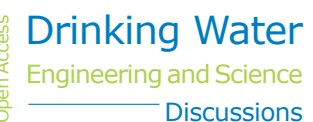

Subsequently, the inflow to the sewerage is predicted. The sum of runoff, DWF and discharge predictions of connected upstream catchments determine the inflow prediction. The runoff is based on the connected paved area and the precipitation prediction. In the third step, the sewerage is modelled as a single reservoir for each catchment, which is optimally used
within the imposed constraints. This results in the optimal outflow prediction. The optimisation technique is in principle equal to the predictive control applied in drinking water supply (Bakker et al. 2013b).

Only if the actual level in the sewerage is lower than a critical level and the predicted precipitation is below a threshold, discharge limitation is turned on. The discharge is limited to an optimal capacity meeting the imposed constraints. In case a
subsequent significant rainfall event is forecasted at a moment before the sewerage can be completely emptied at post-treatment capacity, discharge will be optimised. Depending on the local configuration, the discharge can be either gradually limited between maximum and post-treatment capacity, or kept at maximum capacity for a calculated period after the rainfall event. In this way, the sewerage is emptied before the next rainfall event and bypass of the post-treatment is minimised.

**2.4 Implementation**

For the pilot project this control is implemented for the WWTPs Bennekom, Ede, Woudenberg and Harderwijk (Netherlands). These are conventional WWTPs with a post-treatment. The allocated hydraulic capacity of the post-treatment is for all these locations lower than the hydraulic capacity of the WWTP. The implementation is executed in close cooperation between Water authority Vallei and Veluwe and the connected municipalities. The predictive controller is implemented for all four locations. This study analyses the implementation of WWTP Bennekom in detail.


Before implementation, influent flow at WWTP Bennekom took place at three different stages: 200, 500 and 1000 $m^3$/h. Only switching between fixed stages is possible, since none of the screw pumps are provided with variable speed drives (VSDs). However, above 700 $m^3$/h silting of the sand filters occur. Therefore, an extra fixed stage of 700 $m^3$/h was configured for this project. In case of suitable conditions discharge is limited from 1000 to 700 $m^3$/h by the predictive
control.

**2.5 Phasing and monitoring**

The predictive control was first implemented in advisory mode to check the results in the real-time, full-scale situation. After this period the control is switched on if the results from the advisory period were satisfactory. The control is continuously monitored by the application of key performance indicators (KPIs) for both sewerage and post-treatment. Storage utilisation
and travelling time were defined as main KPIs for the sewerage. The efficiency of the predictive control itself was defined as a major KPI for the post-treatment. The efficiency of the predictive control is determined as the ratio of the bypass of the post-treatment prevented and the amount of bypass for the situation without predictive control (Icke et al. 2016).



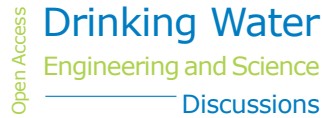

## 3 Results and discussion

In advisory mode the results prove that most of the rainfall events could have been discharged to the WWTP with a reduced capacity and therefore the amount of bypass can be reduced without extra CSO. At the beginning of February 2016 the predictive control for WWTP Bennekom was activated. The predictive control has been continuously activated, apart from the period between the 10th and 12th of February and the 13th and 14th of March due to maintenance activities. During the analysed period (February - September) several precipitation events which could cause bypass occurred.

### 3.1 Performance control total period


The performance of the predictive control at WWTP Bennekom for the analysed period is shown in Figure 2. Both the amount of bypass that occurred and the amount that was prevented are presented for each day. For those days where the predictive control was (partially) deactivated due to maintenance activities the prevented bypass was predicted.

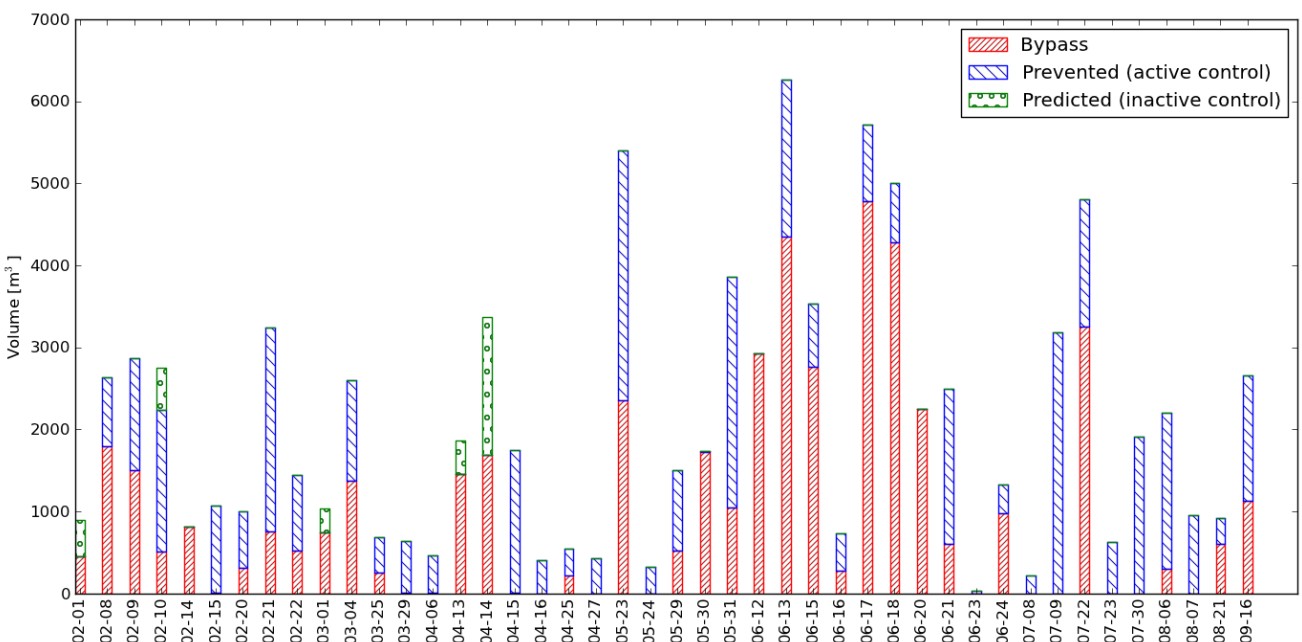


**Figure 2: Performance of the predictive control at WWTP Bennekom for the period February – September 2016.**

For the analysed period (February – September 2016) the results at WWTP Bennekom show that 51% of bypass volume has been prevented with predictive control compared to the original situation with level-based control. The results also show that

big differences occur between the months regarding effect of the predictive control. The analysed period covers ⅔ of the year including the whole summer period.




During the first three months (February – April) 67% of the bypass could have been prevented by predictive control; however this was in reality 12% lower due to maintenance activities. The effects of subsequent rainfall events were limited, since the intervals between distinct predicted events were in general longer than the travelling time with reduced capacity (Icke et al. 2016). Maximum travelling times did not exceed 12 hours and the reduction of bypass occurred without extra CSO. The precipitation during this (early) spring period can be characterised as gradual.

During the next two months (May and June) 33% of the bypass was prevented by predictive control. Especially June was a month with severe thunderstorms. The precipitation during this (early) summer period can be characterised as difficult to predict and large amounts within short time periods. The effects of subsequent rainfall events were large. Discharge limitation of the predictive control can only occur below critical levels and without significant rainfall in the near future. Finally, during the last three months (July – September) 70% of the bypass was prevented by predictive control.

### 3.2 Performance single rainfall events

In this pilot project, a conservative approach is used, in which the predictive control is limited *after* the rainfall event. In Figure 3 bypass reduction *after* a rainfall event of the 4[th] of March is illustrated.

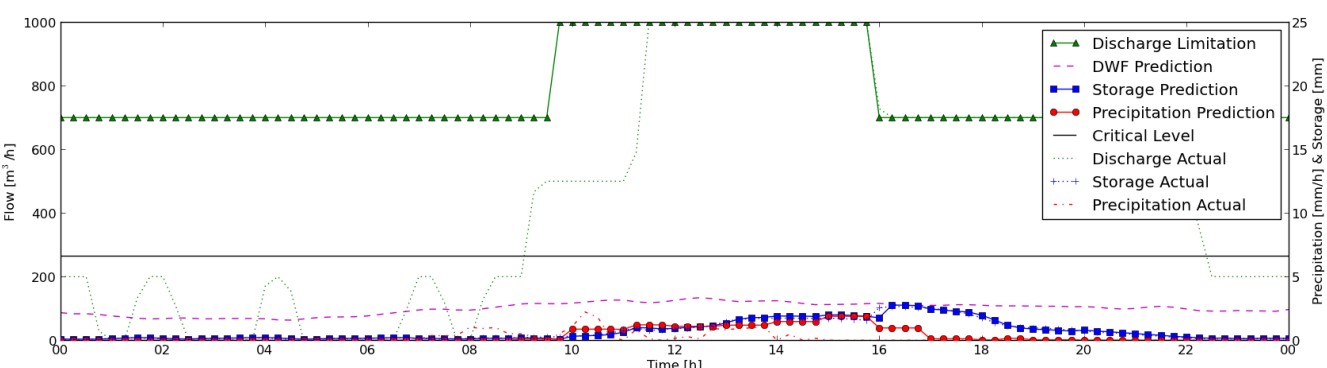

**Figure 3: Example of bypass reduction with a conservative approach at WWTP for the event of the 4[th] of March (Icke et al. 2016).**

The efficiency of the predictive control for this single event accounts 47%. This is entirely due to the chosen approach of applying discharge limitation *after* the rainfall event, not a subsequent rainfall event. In case the progressive approach had been applied, allowing discharge limitation *during* the rainfall, bypass could be completely prevented for this specific event. This holds for the majority of the events in the spring period February – April (Icke et al. 2016).

The progressive approach could also be of interest for the summer period. In Figure 4 bypass reduction *after* a rainfall event of the 16[th] of September is illustrated. Also for this thunderstorm, bypass could be completely prevented with the progressive



approach. Although the progressive approach is riskier regarding CSO as mentioned before, analysis of subsequent events shows that this risk is limited by application of the volume optimisation technique.


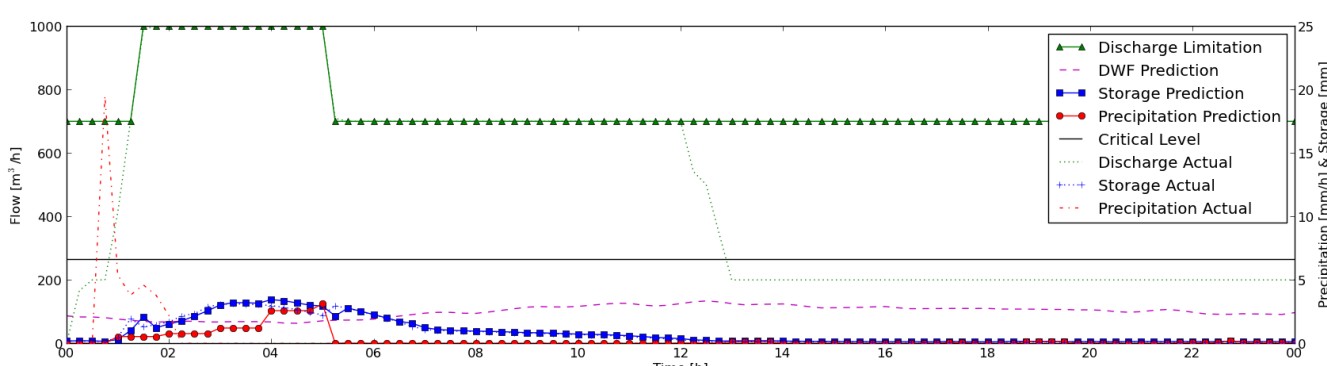

**Figure 4: Example of bypass reduction with a conservative approach at WWTP for the event of the 16<sup>th</sup> of September.**

## 3.3 Precipitation prediction

The results show that most rainfall events are adequately predicted with respect to appearance and timing and it is shown that

this prediction can be used for control. However, further research can show if shape and volume of rainfall forecast might be improved by application of more advanced techniques, or combination of techniques. Especially for the summer period Hirlam Aladin Regional on Mesoscale Operational NWP in Euromed (HARMONIE) precipitation prediction performs better than HIRLAM (Hooijman 2014). In addition, HARMONIE has raster cell sizes of 2,5 by 2,5 km (KNMI 2010), which results in a more distinctive selection of cells for each catchment. So-called nowcasting, with a forecast horizon between 0

and 6 hours, offers more accurate information for the very short-term, but it loses its value for the long-term (Golding 1998). Although the forecast horizon of nowcasting is too short in comparison to travelling times in the sewerage, it might be meaningful to combine it with HARMONIE to use the best of both worlds. Also the usage of the spread of Ensemble forecasts accounting for uncertainty is considered. Discharge limitation could be disabled in case of unpredictable weather.

## 4 Conclusion

Flow intake based on predictive control using DWF and rainfall predictions offers reduced peak discharges on the WWTPs. This results in a better performance of the WWTP and particularly the utilisation of the post-treatment phase, which improves the surface water quality. For the analysed period (February – September 2016) results at WWTP Bennekom show that about 50% of bypass volume of the post-treatment phase can be prevented with operational predictive control. Analysis of single rainfall events showed that the approach, in which the discharge is limited after the rainfall event, is still

conservative. The prevented amount can be even further increased by allowing limited discharge during significant rainfall.



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
