# Peer review of "Flow Intake Control using Dry-weather Forecast"

_Drinking Water Engineering and Science, 2017_

## Referee Comment (RC1) · Anonymous Referee #1 · 13 Feb 2017

In the manuscript entitled "Flow Intake Control using Dry-weather Forecast", the authors propose an interesting real-time approach to improve the wastewater control in case of flooding events. Some main changes should be addressed before to be keen to recommend this work to be accepted in DWES.

I highlight the following:

The Introduction explains properly the peak discharge issues after heavy rainfall events. However, I miss a clear exposition of what the authors are proposing to do in the paper.

In Section 2 the authors introduce Aquasuite(R). This is well referenced but it would be of interest for the reader if the authors also provide a website where find further information, in case it is available on-line.

In page 3, line 70, the authors claim: "This particular technique is based on the fully

adaptive forecasting model for short-term drinking water demand (Bakker et al. 2013a) applied on waste water discharge." This point should be clarified about if the method used is general enough or how it is adapted in this case to also cover the analysis of both drinking and wastewater. The subsequent 2 sentences don't clarify this point. Please, explain.

At line 81, the authors should reword the paragraph to mean that the 3 phases explanation properly corresponds to the initial statement of "a volume optimisation technique..." which seems quite generic

At line 91, the sentence "The optimisation technique is in principle equal to the predictive control applied in drinking water supply (Bakker et al. 2013b)" is confuse and should be explained.

Subsection 2.4 seems more a case-study introduction than a proper implementation as it is suggested by the title. However, as a case-study is short of details as well as describing the working implementation. The authors should reword this subsection and provide more and better details.

At line 174: "So-called nowcasting..." The authors should explain better what is nowcasting and it is a widely used and hugely important topic. The authors should also be more specific in how the current approach

As a side note, it is worth mentioning that the authors should reduce as much as they can the use of "past participles" in the text.

---

## Short Comment (SC1) · 19 May 2017

The paper describes a nice and interesting study on the use of an operational real-time control system. As these systems are rarely actually build and operationally used, this is interesting on beforehand.

I have only a few questions and remarks:

In Section 1 it is stated that there is a "significant bypass of post-treatment during peak discharge". Please elaborate this with illustrative figures, for example in terms of spilled load or volumes to surface water relative to total. This, to get an idea of impact and potential benefits.

You refer a few times to optimization of inflow "by using predictive control", using predictions on inflow. But, what exactly are you controlling and how? I suspect by controlling the sewerage pumping stations, which are used to pump the waste water from sewer

systems into the transport pipe system? Or…?

Line 35: "To verify the prelimary…" should be "To verify the results of the prelimary…." ?

With respect to section 2.1: is there anything to say about the forecast accuracy of the HiRLAM NWP, especially in terms of precipitation depths? Why (old) HiRLAM is chosen and not the newer and (probably better and more accurate) HARMONIE forecast? What about using uncertainty techniques, using for example ensembles? There is some discussion in 3.3, but why you chose Hirlam and not harmonie on beforehand?

Section 3: I presume that the bypass volume is observed, while the prevented volumes are based on simulations?

Lines 155-160: you state that bypass could completely be prevented. What about the possibly increased CSO volume in this case?

Line 163-164: completely unclear to me what you mean here.

Maybe a little out of scope for this paper, but given the fact that these kind of predictive control systems are not yet common practise, do you have any additional experiences, do's and don'ts which can be shared with us?

Please also note the supplement to this comment:
http://www.drink-water-eng-sci-discuss.net/dwes-2017-1/dwes-2017-1-SC1-supplement.pdf

---

## Author Comment (AC1) · 19 Jun 2017

*First of all, thanks for your useful questions and remarks! We appreciate your comments! Answers to the questions and remarks of Anonymous Referee #1:*

1. The Introduction explains properly the peak discharge issues after heavy rainfall events. However, I miss a clear exposition of what the authors are proposing to do in the paper.

   *The introduction will be expanded with a brief preview of what the reader can expect in this paper. In this way, the reader will be better guided.*

2. In Section 2 the authors introduce Aquasuite®. This is well referenced but it would be of interest for the reader if the authors also provide a website where to

find further information, in case it is available on-line.

*The URL is mentioned in the reference where further information about Aquasuite® can be found: http://aquasuite.net/*

3. In page 3, line 70, the authors claim: "This particular technique is based on the fully adaptive forecasting model for short-term drinking water demand (Bakker et al. 2013a) applied on waste water discharge." This point should be clarified about if the method used is general enough or how it is adapted in this case to also cover the analysis of both drinking and wastewater. The subsequent 2 sentences don't clarify this point. Please, explain.

*This technique is so generic that it can be applied on waste water discharge with some small adjustments of the settings. This method has been successfully applied for several WWTPs. Not only for predictive control on quantity but also on quality. This particular paragraph will be modified.*

4. At line 81, the authors should reword the paragraph to mean that the 3 phases explanation properly corresponds to the initial statement of "a volume optimisation technique..." which seems quite generic.

*We will reword this paragraph in combination with the following paragraph so that the structure of the text becomes clearer.*

5. At line 91, the sentence "The optimisation technique is in principle equal to the predictive control applied in drinking water supply (Bakker et al. 2013b)" is confusing and should be explained.

*This sentence will be further explained. In drinking water supply this technique is applied to flatten the consumption or distribution pattern with a reservoir*

*to obtain an as flat as possible production pattern. In this particular study, this technique is applied to waste water discharge to optimise the flow to the treatment plant using the available storage of the sewer after precipitation events.*

6. Subsection 2.4 seems more a case-study introduction than a proper implementation as it is suggested by the title. However, as a case-study is short of details as well as describing the working implementation. The authors should reword this subsection and provide more and better details.

   *It was chosen to briefly describe the implementation of the predictive controller without too much consideration of the details of the supporting process automation layer. It was chosen to amplify the subsection implementation with the subsection phasing and monitoring.*

7. At line 174: "So-called nowcasting..." The authors should explain better what is nowcasting and it is a widely used and hugely important topic.

   *We will modify the text here and give a better explanation of the principle of nowcasting and its possible meaning for predictive control.*

8. As a side note, it is worth mentioning that the authors should reduce as much as they can the use of "past participles" in the text.

   *We will check and scan the grammar in the paper and the construction of the sentences.*

---

## Author Comment (AC2) · 19 Jun 2017

*First of all, thanks for your useful questions and remarks! We appreciate your comments! Answers to the questions and remarks of referee Klaas-Jan van Heeringen:*

1. In Section 1 it is stated that there is a "significant bypass of post-treatment during peak discharge". Please elaborate this with illustrative figures, for example in terms of spilled load or volumes to surface water relative to total. This, to get an idea of impact and potential benefits.

   *For this paper, we primarily focused our research on the efficiency of the predictive controller reducing the amount of bypass in relation to the capacity of the post-treatment. Nevertheless, your suggestion for a quantitative analysis on the impact of the volumes and loads on the surface water is interesting for*

[Figure]

*further research in the future. For now we would like to keep the scope of this paper in mind.*

2. You refer a few times to optimization of inflow "by using predictive control", using predictions on inflow. But, what exactly are you controlling and how? I suspect by controlling the sewerage pumping stations, which are used to pump the waste water from sewer systems into the transport pipe system? Or...?

   *The predictive control is applied on the influent flow to the Waste Water Treatment Plant (WWTP). So this can be the influent pumping station at the WTTP or the sewerage pumping stations of the water board for catchment areas discharging directly to the WWTP. It depends on the exact configuration of catchment areas and pumping stations. For the case of WWTP Bennekom it concerns the influent pumping station (screw pumps) at the WWTP. We will clarify this in the paper.*

3. Line 35: "To verify the preliminary..." should be "To verify the results of the preliminary..." ?

   *We totally agree with that. To be corrected in the next version of this paper.*

4. With respect to section 2.1: is there anything to say about the forecast accuracy of the HiRLAM NWP, especially in terms of precipitation depths? Why (old) HiRLAM is chosen and not the newer and (probably better and more accurate) HARMONIE forecast? What about using uncertainty techniques, using for example ensembles? There is some discussion in 3.3, but why you chose Hirlam and not Harmonie on beforehand?

   *At the beginning of the project, HARMONIE could not be implemented due*

*to practical limitations. HIRLAM, HARMONIE and Ensemble forecasts were discussed in the project group. It was decided to start with HIRLAM due to the practical limitations. On the one hand, the study of [Hooijman 2014] showed that HARMONIE better performs than HIRLAM especially for the summer period. On the other hand, investigation of all single events during the period February 2016 – January 2017 for our study area Bennekom showed that for only one precipitation event the prediction of HIRLAM really lacked accuracy. Replacement of HARMONIE with HIRLAM and the application of Ensemble forecasts and nowcasting are again on the agenda of the project group to be discussed.*

5. Section 3: I presume that the bypass volume is observed, while the prevented volumes are based on simulations?

   *The bypass volume is indeed based on measurements, whereas the prevented volumes are based on real-time concurrent calculations of what would happen if discharge was not limited.*

6. Lines 155-160: You state that bypass could completely be prevented. What about the possibly increased CSO volume in this case?

   *Analysis of the events of this specific period (early spring) shows that the current control is still quite conservative due to the chosen approach of applying discharge limitation after the rainfall event. Actually, applying discharge limitation during the rainfall event would not have led to increased CSO. The precipitation prediction appears accurate enough during the winter to apply discharge limitation based on the cumulative volume prediction without other restrictions.*

7. Line 163-164: completely unclear to me what you mean here.

*This sentence will be rewritten. This sentence intended to summarize that the progressive approach is worth considering although this is riskier regarding CSO. By applying only the volume optimisation technique without any restrictions, the accuracy of the precipitation prediction becomes more important.*

8. Maybe a little out of scope for this paper, but given the fact that these kind of predictive control systems are not yet common practise, do you have any additional experiences, do's and don'ts which can be shared with us?

*We have additional experiences with predictive control systems especially for the production and distribution of water supply systems, but also for the aeration and return sludge of waste water treatment plants. Indeed, this might be a little out of scope for this paper. Nevertheless, in this project we experienced that predictive control with a robust and straightforward model delivers satisfactory results. Also the ability to easily couple with several real-time data sources is a must. Important is to investigate the implications of the configuration of the distinct sewer areas and pumping stations on the several possibilities for optimisation.*